# Value of C-11 methionine PET/CT in patients with intracranial germinoma

Yong-Jin Park[1], Ji Won Lee[2], Hee Won Cho[2], Yearn Seong Choe[3], Kyung-Han Lee[3], Joon Young Choi[3], Ki Woong Sung[2]ᴼ*, Seung Hwan Moon[3]ᴼ*

**1** Department of Nuclear Medicine, Soonchunhyang University Cheonan Hospital, Cheonan, Chungcheongnam-do, South Korea, **2** Department of Pediatrics, Samsung Medical Center, Sungkyunkwan University School of Medicine, Seoul, South Korea, **3** Department of Nuclear Medicine, Samsung Medical Center, Sungkyunkwan University School of Medicine, Seoul, South Korea

ᴼ These authors contributed equally to this work.
* kiwoong.sung@samsung.com (KWS); seunghwan.moons.moon@samsung.com (SHM)

## Abstract

### Purpose

The purpose of this study was to investigate the value of C-11 methionine (MET) positron emission tomography (PET)/computed tomography (CT) in patients with intracranial germinoma (IG).

### Material and methods

We conducted a retrospective analysis of 21 consecutive patients with pathologically confirmed IGs and eight patients with intracranial non-germinomas (INGs) located in a similar region. Clinical characteristics, imaging findings, and tumor markers such as α-fetoprotein (AFP) and β-human chorionic gonadotropin (HCG) were used as clinical variables. Maximum standardized uptake value ($SUV_{max}$), tumor-to-normal tissue (T/N) ratio, and visual scoring of tumor were used as MET PET parameters.

### Results

All IGs were well visualized on MET PET with a three-grade visual scoring system. In addition, $SUV_{max}$ of IGs was higher than that of INGs (P = 0.005). Pre-treatment (Pre-Tx) T/N ratio was significantly correlated with pre-Tx serum HCG (P = 0.031). Moreover, MET PET parameters showed significant associations with tumor location, sex, *KRAS* variant, and symptoms.

### Conclusion

MET PET/CT could be a useful diagnostic tool in patients suspected of having IGs. In addition, the MET avidity of tumor is a potential surrogate biomarker of HCG, which has been used as a diagnostic marker for IGs. Tumor MET parameters also had significant differences according to tumor locations, sex, symptoms, and *KRAS* mutation. However, MET avidity of tumors had no significant prognostic value.

**Data Availability Statement:** All relevant data are within the manuscript and its Supporting Information files.

**Funding:** This work was supported by the National Research Foundation of Korea(NRF) grant funded

by the Korea government(MSIP) (No.NRF-2019R1F1A1060353) (SHM) and the Soonchunhyang University Cheonan Hospital Research Fund (YJP). The funders had no role in study design, data collection and analysis, decision to publish, or preparation of the manuscript.

**Competing interests:** The authors have declared that no competing interests exist.

## Introduction

Intracranial germinomas (IGs) originate from totipotent germ cells and usually occur in children and young adults [1, 2]. They are the most common type of intracranial germ cell tumors (GCTs) and account for 0.5–2.1% of all pediatric primary brain tumors [3]. IGs are potentially malignant in behavior, and can infiltrate normal brain tissue as well as spread through the whole of the central nervous system (CNS) [4]. However, these lesions are highly sensitive to radiotherapy or/and chemotherapy, and are potentially curable without complete surgical resection [5–7]. Given that these tumors have a highly favorable treatment response and delayed treatment can cause more severe neurologic complications, early and accurate diagnosis is essential for better clinical outcomes in patients with IGs. Clinical diagnosis and staging of IGs are based on symptoms, tumor markers such as α-fetoprotein (AFP) or β-human chorionic gonadotropin (HCG) in serum and cerebrospinal fluid (CSF), and brain and spinal magnetic resonance imaging (MRI) [8]. AFP or HCG values beyond a certain threshold in either the serum or cerebrospinal fluid suggest the presence of these specific tumors. However, on the basis of experience from previous clinical studies, the defined thresholds can vary according to clinical scenario [8]. MRI is more sensitive than computed tomography (CT) and thus is considered the imaging modality of choice. However, in some cases of basal ganglia germinomas, it was difficult to detect the tumor lesions because the lesions in basal ganglia only show subtle signal change, no mass effects, and nonspecific radiological findings on MRI during their early stages [7, 9–11]. Therefore, an additional biomarker reflecting the tumor characteristics would improve diagnosis and management of these tumors.

C-11 methionine (MET) positron emission tomography (PET)/CT has potential as an additional imaging tool for brain tumors [12]. Methionine easily crosses the intact blood-brain barrier via the neutral amino acid transporter and is incorporated into active tumor [10]. In a previous study on the distribution of MET in mouse malignant tumor models, MET uptake was mainly by viable cancer cells, and MET uptake by macrophages and granulation tissues was low [13]. Previous studies and case reports in patients with IG using MET PET have mainly done research on basal ganglia germinomas [3, 6, 7, 10, 11, 14], and described that MET PET is useful for early detection of basal ganglia germinoma without overt mass formation, assessment of treatment response and residual tumor, and monitoring for tumor recurrence [3, 6, 7, 10]. In previous studies, patients with basal ganglia germinoma presented only with hemiatrophy without any obvious mass formation on MRI, and the studies explained that MET PET was valuable for detecting the precise location of the tumor [10, 11]. In another previous study in IGs, it elucidated the characteristic MET and F-18 fluorodeoxyglucose (FDG) PET findings in 10 patients with IGs and reported that MET is more useful for tumor contouring and treatment planning because MET has better image contrast than FDG [9]. However, due to the rarity of IG, the value of MET PET/CT for this disease has not yet been fully investigated.

In the present study, we evaluated the value of MET PET/CT in patients with IG. We investigated associations between MET PET parameters and clinical variables such as tumor markers, tumor location, tumor size, symptoms, and other clinical factors. In addition, the MET PET findings of IGs and intracranial non-germinoma (INGs) that develop in pineal region, sellar-suprasellar region, and basal ganglia were compared, and the prognosis of IG was also investigated.

## Materials and methods

### Patient selection

We reviewed the electronic medical records of 1,685 patients who underwent MET PET/CT at Samsung Medical Center from December 2013 to May 2020. Among these patients, 66

consecutive patients with pathologically confirmed IG were first identified. From a total of 66 patients, 45 patients who did not undergo MET PET/CT before or after treatment were excluded. Finally, 21 patients were enrolled in this study, and pre-treatment (pre-Tx) and post-treatment (post-Tx) MET PET/CT images from the 21 patients were used for analysis. Of the 21 patients, 20 patients had one IG, while one patient had two IGs, located in pineal region and sellar-suprasellar region.

To compare the MET uptake of IGs with INGs, we searched the electronic medical records to identify pathologically confirmed INGs in the same region during the same study period. Eight patients with INGs located in pineal region, sellar-suprasellar region, or basal ganglia were this study. Final analysis included 22 IG lesions in 21 patients and eight ING lesions in eight patients.

The present study was approved by the institutional review board (2020-11-060-002). The need for informed consent was waived because of the retrospective nature of the study and the anonymous clinical data were analyzed. All processes performed in this study involving human participants were in accordance with the ethical standards of institutional review board of our medical center and with the declaration of Helsinki 1964 and its later amend-ments or comparable ethical standards.

## C-11 methionine PET/CT scanning protocol

Study patients fasted for at least 4 hours, and then an intravenous injection of MET was given. The injection dose of MET was 6.0 MBq/kg (162 μCi/kg). When pediatric patients needed to be sedated, their vital signs were monitored during sedation. After 20 minutes of MET injec-tion, patients underwent MET PET/CT scan using a Discovery STE PET/CT scanner (GE Healthcare, Waukesha, WI, USA). The entire brain was imaged. CT scans were performed via continuous spiral method using a 16-slice helical CT, and acquisition parameters of CT images were 120 kVp, 50 mA, rotation time 0.8 second, helical thickness 3.75 mm, helical interval 3.27 mm, pitch 1.75:1, and beam collimation 10 mm. After performing CT scans, PET scans were performed for 7 minutes/frame, and PET images were reconstructed using an iterative three-dimensional reconstruction method (20 subsets, 2 iterations). Acquisition parameters of PET images were $128 \times 128 \times 48$ matrix with 2 mm × 2 mm × 3.27 mm voxel size. Standardized uptake values (SUVs) were calculated by adjusting for the injection dose of MET and body weight of patient. Advantage Workstation VolumeShare 7 (AW 4.7, GE Healthcare) was used for co-registration of CT and PET images.

## Analysis of C-11 methionine PET/CT images

Advantage Workstation VolumeShare 7 (AW 4.7, GE Healthcare) software was used to mea-sure PET parameters. In patients with IG, the mean durations between pre-Tx MET PET/CT and the start of treatment were 8.5 days (median: 1 day), and the mean durations between the end of treatment and post-Tx MET PET/CT were 12.1 months (median: 13 months). Of 22 IGs, 17 (77.3%) were biopsy performed prior to pre-Tx MET PET/CT, while the remaining five (22.7%) were pre-Tx MET PET/CT followed by biopsy. Volumes of interest (VOIs) were manually drawn by two experienced nuclear medicine physicians on MET PET/CT images to delineate the exact contours of IGs with reference to non-contrast CT images of PET/CT and contrast images of MRI. Maximum SUV ($SUV_{max}$) and tumor-to-normal tissue (T/N) ratio on pre-Tx and post-Tx MET PET/CT images were measured. The T/N ratio of IG was defined as $SUV_{max}$ of IG divided by mean SUV ($SUV_{mean}$) of normal frontal cortex [9]. The $SUV_{mean}$ of the normal frontal cortex was measured using spherical VOI.

Pre-Tx and post-Tx MET uptake of IGs was evaluated by two experienced nuclear medicine physicians using a three-grade visual scoring system. Grade 1 (negative) indicated that MET

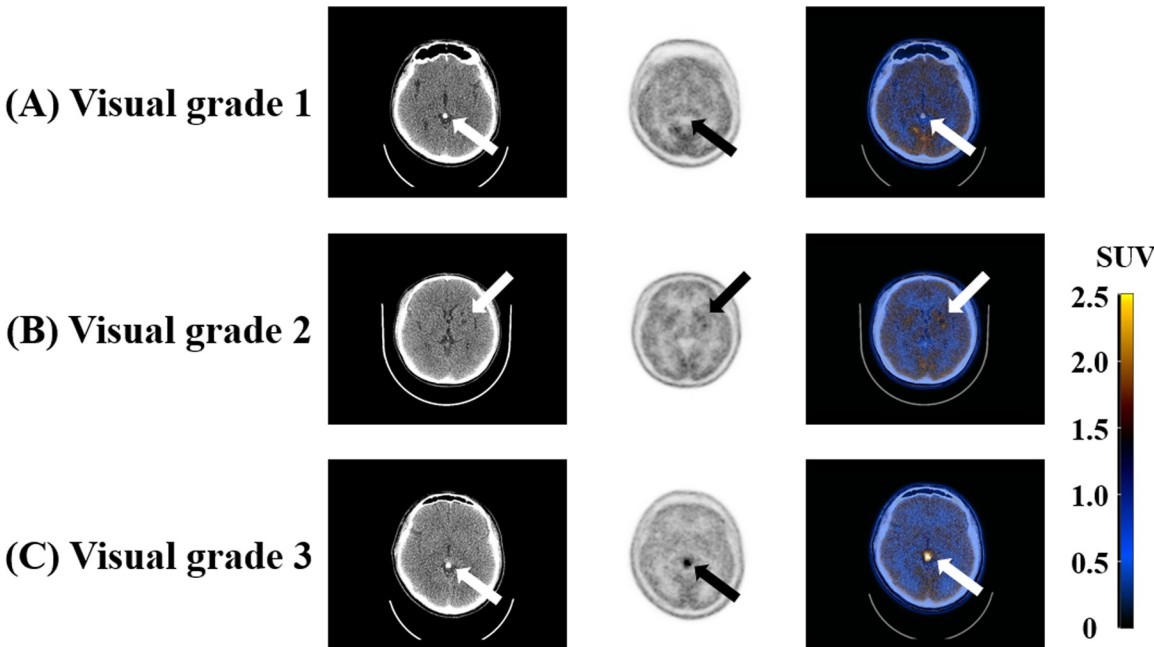

**Fig 1. Examples of the three-grade visual scoring system for IG.** (A) A case of visual grade 1 was a 24-year-old male patient with pineal germinoma and was post-Tx MET PET/CT images. Tumor size was 9 mm, and post-Tx $SUV_{max}$ and T/N ratio were 1.52 and 1.10, respectively. The MET uptake of tumor was indistinguishable from that of normal brain tissue. (B) A case of visual grade 2 was a 13-year-old male patient with left basal ganglia germinoma and was post-Tx MET PET/CT images. Tumor size was 12 mm, and post-Tx $SUV_{max}$ and T/N ratio were 2.26 and 1.85, respectively. The MET uptake of tumor showed slightly higher MET uptake than that of surrounding normal brain tissue. (C) A case of visual grade 3 was 23-year-old male patient with pineal germinoma and was pre-Tx MET PET/CT images. Tumor size was 19 mm, and pre-Tx $SUV_{max}$ and T/N ratio were 3.97 and 3.45, respectively. The MET uptake of tumor was distinctly higher than that of surrounding normal brain tissue. IG, intracranial germinoma; post-Tx, post-treatment; MET, C-11 methionine; PET, positron emission tomography; CT, computed tomography; $SUV_{max}$, maximum standardized uptake value; T/N, tumor-to-normal tissue; pre-Tx, pre-treatment; SUV, standardized uptake value.

uptake was indistinguishable from normal brain tissue, and grade 3 (positive) indicated that MET uptake was markedly higher than background uptake and clearly distinguished from background uptake. Grade 2 (weakly positive) indicated that MET uptake was neither grade 1 (negative) nor grade 3 (positive). Therefore, in the visual grade 2, the MET uptake was slightly higher than background uptake and vaguely distinguished from the background uptake. Examples of the visual scoring system are shown in Fig 1. Enhancements in IG were investigated on conventional CT and MRI, and calcifications in IGs were investigated on conventional CT.

In this study, 22 IGs were located in the pineal region, sellar-suprasellar region, and basal ganglia. During the same study period, eight pathologically confirmed INGs in the pineal region, sellar-suprasellar region, and basal ganglia were enrolled and analyzed. The eight INGs included two cavernous angiomas, one glioneuronal tumor, one low-grade astrocytic tumor, one low-grade glioma, one immature teratoma, one langerhans cell histiocytosis, and one mixed germ cell tumor. The differences in the $SUV_{max}$ and T/N ratio of the 22 IGs and eight INGs were investigated. Tumor locations, $SUV_{max}$, and T/N ratio of eight INGs are presented in S1 Table.

## Clinical variables and PET parameters

In the present study, age, sex, leptomeningeal seeding, symptom variables, and tumor markers were clinical variables. The symptom variables consisted of headache, nausea, vomiting,

dizziness, visual impairment, polydipsia, polyuria, short stature, one-sided weakness, clumsiness, dysarthria, weight loss, and xerostomia. The tumor marker variables consisted of pre-Tx CSF AFP, pre-Tx serum AFP, pre-Tx CSF HCG, pre-Tx serum HCG, post-Tx CSF AFP, post-Tx serum AFP, post-Tx CSF HCG, post-Tx serum HCG, Δ CSF AFP, Δ serum AFP, Δ CSF HCG, and Δ serum HCG. The difference in the variables pre-Tx and post-Tx was described as Δ in this study. The tumor markers were measured within three days of MET PET/CT scans. In addition, tumor location, tumor size, enhancement on CT and MRI, and calcification on CT were also clinical variables. Tumor locations consisted of the pineal region, sellar-suprasellar region, and basal ganglia.

The associations of PET parameters with clinical variables were also analyzed. PET parameters consisted of pre-Tx T/N ratio, pre-Tx $SUV_{max}$, post-Tx T/N ratio, post-Tx $SUV_{max}$, Δ T/N ratio, and Δ $SUV_{max}$. Clinical variables and PET parameters of IG are presented in Tables 1 and 2.

In addition, eight (38.1%) of 21 patients with IGs underwent next-generation sequencing (NGS), and 11 gene variants were discovered: *KIT*, *KRAS*, *NRAS*, *TERT*, *APC*, *PTPRD*, *KMT2C*, *KMT2D*, *RUNX1*, *PTCH1*, and *PIK3CA*, and the association between these 11 genetic variants and PET parameters was also investigated.

## Statistical analysis

IBM SPSS statistics software version 27.0 for Windows (IBM Corporation. Armonk, NY, USA) was used for analysis in this study. Independent T-test and Mann-Whitney U test were used to compare the continuous variables between two groups. When comparing continuous variables among three groups, Kruskal-Wallis test was employed because continuous variables did not have a normal distribution. In post hoc testing, Mann-Whitney U test with Bonferroni's method was used to compare the PET parameters of two different tumor locations, and a significant difference was defined as a P value less than 0.016. The correlation of two continuous variables was analyzed using Pearson correlation and Spearman correlation.

The prognostic significance of PET parameters and other clinical variables related to progression-free survival (PFS) and overall survival (OS) were assessed by univariate and multivariate analyses using Cox proportional hazards regression models with enter method. PFS was defined as the time from the date of chemotherapy initiation to the date of documented progression or last clinical follow-up. OS was defined as the time from the date of chemotherapy initiation to cancer or treatment-related death or last clinical follow-up.

Using graphs of MedCalc statistical software version 20.009 for Windows (MedCalc software, Ostend, Belgium), comparisons of pre-Tx $SUV_{max}$ and pre-Tx T/N ratio between 22 IGs and eight INGs located in pineal region, sellar-suprasellar region, and basal ganglia were presented. In addition, comparisons of pre-Tx $SUV_{max}$ and pre-Tx T/N ratio between IGs located in the pineal region, sellar-suprasellar region, and basal ganglia were presented.

## Results

### Characteristics of 21 patients with 22 IGs

A total of 21 patients with IG were enrolled. The patient group consisted of 15 men (71.4%) and 6 women (28.6%), and the median age of patients was 16.0 years. Of the 21 patients, one (4.8%) relapsed after treatment and died undergoing additional surgical treatment, chemotherapy, and radiation therapy. The other 20 (95.2%) did not experience recurrence after chemotherapy and radiation therapy. All patients were managed by the same chemotherapy regimen and subsequent radiotherapy as initial treatment, and one patient who relapsed was given a changed treatment regimen after the relapse. Five (23.8%) of 21 patients had

leptomeningeal seeding at the time of initial diagnosis. A total of 21 patients complained of 10 symptoms during initial diagnosis before treatment. Seven (33.3%) had polydipsia and poly-uria, and six (28.6%) had headache. Five (23.8%) had visual impairment, and four (19.0%) had nausea, vomiting, and dizziness. Three (14.3%) had short stature and one-sided weakness, and one (4.8%) had dysarthria, weight loss, and xerostomia. Median pre-Tx serum AFP, CSF AFP, serum HCG, and CSF HCG were 1.2 IU/mL, 0.5 IU/mL, 1.2 mIU/mL, and 3.7 mIU/mL, respectively. After treatment, AFP and HCG showed a decreasing tendency. Characteristics of 21 patients with IG are presented in Table 1.

Of the 22 IGs, nine (40.9%) were located in the pineal region. Ten (45.5%) were located in the sellar-suprasellar region, and three (13.6%) were located in the basal ganglia. The mean size of IGs was 21.1 mm, ranging from 8.0 mm to 44.0 mm. All IGs showed enhancement on CT and MRI, and 13 (59.1%) exhibited calcification on CT. Before treatment, all IGs were

**Table 1. Characteristics of 21 patients with IG.**

|  |  | Number of patients (%) | Median (range) |
|---|---|---|---|
| Age (years) |  |  | 16.0 (6–32) |
| Sex | Men/women | 15/6 (71.4/28.6) |  |
| Recurrence |  | 1 (4.8) |  |
| Death |  | 1 (4.8) |  |
| PFS (months) |  |  | 31.0 (0–77) |
| OS (months) |  |  | 31.0 (0–77) |
| Treatment | CTx + RTx | 20 (95.2) |  |
|  | CTx + RTx + OP | 1 (4.8) |  |
| Leptomeningeal seeding |  | 5 (23.8) |  |
| Symptoms | Polydipsia, polyuria | 7 (33.3) |  |
|  | Headache | 6 (28.6) |  |
|  | Visual impairment | 5 (23.8) |  |
|  | Nausea, vomiting | 4 (19.0) |  |
|  | Dizziness | 4 (19.0) |  |
|  | Short stature | 3 (14.3) |  |
|  | One-sided weakness, clumsiness | 3 (14.3) |  |
|  | Dysarthria | 1 (4.8) |  |
|  | Weight loss | 1 (4.8) |  |
|  | Xerostomia | 1 (4.8) |  |
| Tumor markers | Pre-Tx serum AFP (IU/mL) |  | 1.2 (0.5–5.7) |
|  | Post-Tx serum AFP (IU/mL) |  | 0.7 (0.5–5.8) |
|  | Δ serum AFP (IU/mL) |  | 0.5 (0–2.9) |
|  | Pre-Tx CSF AFP (IU/mL) |  | 0.5 (0.5–1.5) |
|  | Post-Tx CSF AFP (IU/mL) |  | 1.3 (0.3–0.8) |
|  | Δ CSF AFP (IU/mL) |  | 0.0 (0–1.2) |
|  | Pre-Tx serum HCG (mIU/mL) |  | 1.2 (0.4–36.9) |
|  | Post-Tx serum HCG (mIU/mL) |  | 0.7 (0.4–12.1) |
|  | Δ serum HCG (mIU/mL) |  | 0.7 (0–36.5) |
|  | Pre-Tx CSF HCG (mIU/mL) |  | 3.7 (0.6–220.9) |
|  | Post-Tx CSF HCG (mIU/mL) |  | 1.3 (0.4–2.8) |
|  | Δ CSF HCG (mIU/mL) |  | 2.0 (0–219.4) |

IG, intracranial germinoma; PFS, progression-free survival; OS, overall survival; CTx, chemotherapy; RTx, radiotherapy; OP, operation; pre-Tx, pre-treatment; AFP, α-fetoprotein; post-Tx, post-treatment; Δ, difference between pre-treatment and post-treatment; CSF, cerebrospinal fluid; HCG, β-human chorionic gonadotropin.

**Table 2. Characteristics of 22 pathologically confirmed IGs.**

| | | Number of patients (%) | Median (range) |
|---|---|---|---|
| Location | Pineal region | 9 (40.9) | |
| | Sellar-suprasellar region | 10 (45.5) | |
| | Basal ganglia | 3 (13.6) | |
| Tumor size (mm) | | | 18.0 (8.0–44.0) |
| Enhancement on CT and MRI | | 22 (100) | |
| Calcification on CT | | 13 (59.1) | |
| Visual grade | Pre-Tx—1 | 0 (0) | |
| | Pre-Tx—2 | 0 (0) | |
| | Pre-Tx—3 | 22 (100) | |
| | Post-Tx—1 | 16 (72.7) | |
| | Post-Tx—2 | 6 (27.3) | |
| | Post-Tx—3 | 0 (0) | |
| PET parameter | Pre-Tx T/N ratio | | 2.8 (1.6–5.2) |
| | Post-Tx T/N ratio | | 1.1 (0.6–2.1) |
| | Δ T/N ratio | | 1.8 (0–4.2) |
| | Pre-Tx $SUV_{max}$ | | 3.5 (2.1–7.6) |
| | Post-Tx $SUV_{max}$ | | 1.7 (0.8–2.6) |
| | Δ $SUV_{max}$ | | 2.1 (0–5.8) |

IG, intracranial germinoma; CT, computed tomography; MRI, magnetic resonance imaging; pre-Tx, pre-treatment; post-Tx, post-treatment; PET, positron emission tomography; T/N, tumor-to-normal tissue; Δ, difference between pre-treatment and post-treatment; $SUV_{max}$, maximum standardized uptake value.

visual grade 3. After treatment, 16 (72.7%) were visual grade 1, and six (27.3%) were visual grade 2. Mean pre-Tx T/N ratio was 3.0, and mean post-Tx T/N ratio was 1.2 (Table 2). Two representative cases in which the MET uptake of IG was significantly decreased after treatment are shown in Fig 2.

## MET uptake of IGs

All IGs were well visualized on MET PET in this study, as indicated by their visual grade 3 score on pre-Tx scan. In addition, $SUV_{max}$ of IGs were significantly higher than that of INGs (P = 0.005). T/N ratio of IG showed a higher tendency than that of ING, but no statistically significant differences were found (P = 0.222) (S1 Fig).

## Significant correlations between tumor MET uptake and tumor markers

Significant correlations between PET parameters and HCG were identified in this study (Table 3). Pre-Tx T/N ratio was significantly correlated with pre-Tx serum HCG (P = 0.031). Δ T/N ratio was significantly correlated with Δ serum HCG (P = 0.014) and Δ CSF HCG (P = 0.010). Δ $SUV_{max}$ also showed a significant correlation with Δ CSF HCG (P = 0.048). However, no significant correlation was observed between PET parameters and AFP.

## Tumor MET uptake and clinical variables

In the present study, significant differences were identified between PET parameters according to tumor location. There were significant differences in pre-Tx $SUV_{max}$ (P = 0.021) and pre-Tx T/N ratio (P = 0.037) depending on tumor location (Fig 3). On post hoc analysis using Mann-Whitney U test with Bonferroni's method, significant differences in PET parameters were

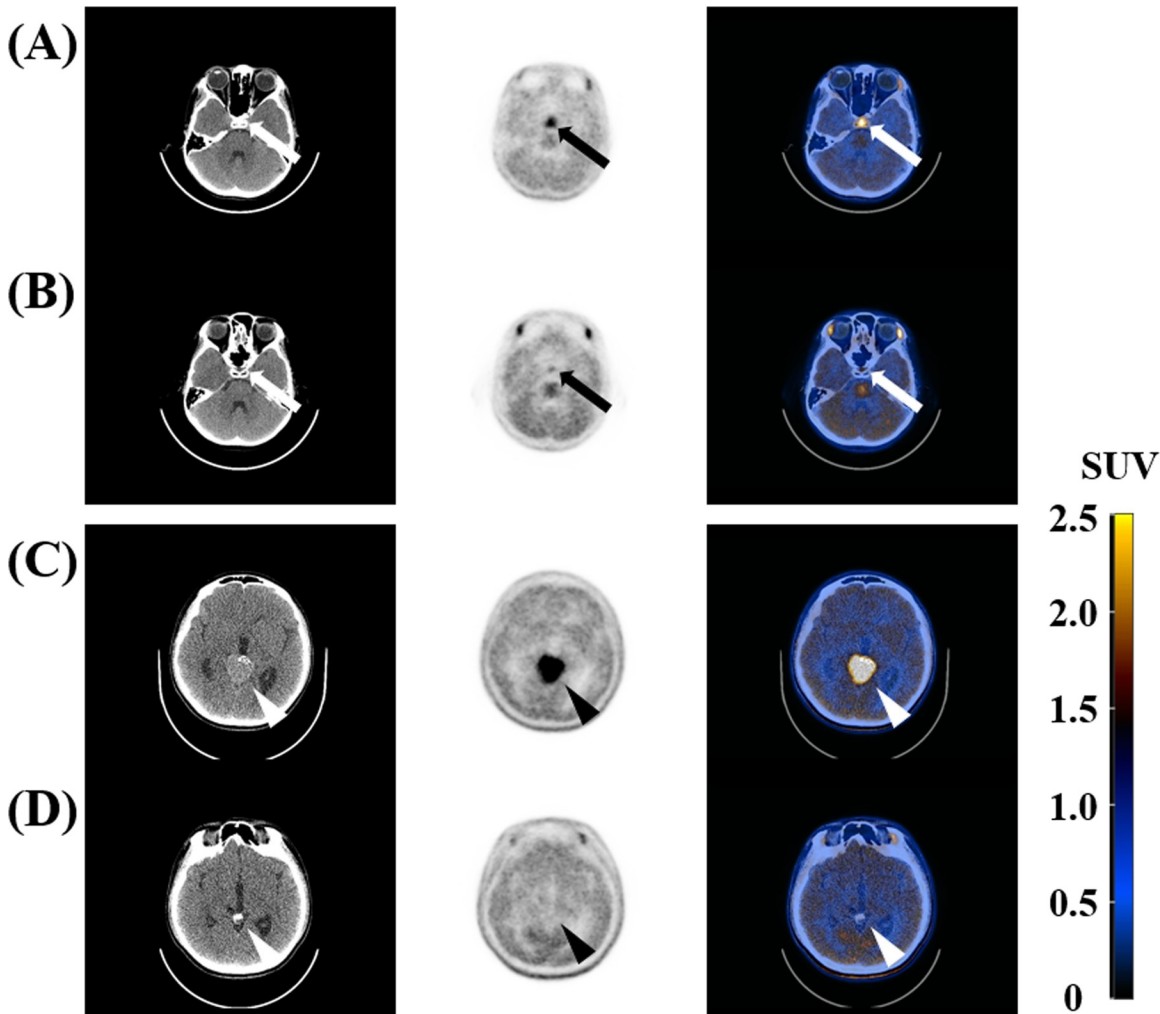

**Fig 2. Non-contrast CT, MET PET, and fusion images of two representative cases in patients with IG before and after treatment.**
(A, B) The first case was an 11-year-old boy, and the patient experienced headache, polydipsia, polyuria, and growth hormone deficiency before treatment. The primary tumor was located in the sellar-suprasellar region (white and black arrows). Before treatment, pre-Tx $SUV_{max}$ and pre-Tx T/N ratio were 4.91 and 4.38, respectively. After chemotherapy and radiation therapy, post-Tx $SUV_{max}$ and post-Tx T/N ratio decreased to 1.62 and 1.27, respectively. The visual grade was three before the treatment and decreased to two after treatment. (C, D) The second case was a 14-year-old boy who experienced headache before treatment. The primary tumor was located in the pineal region (white and black arrow heads). Before treatment, pre-Tx $SUV_{max}$ and pre-Tx T/N ratio were 4.95 and 3.96, respectively. After chemotherapy and radiation therapy, post-Tx $SUV_{max}$ and post-Tx T/N ratio decreased to 1.52 and 1.15, respectively. The visual grade was three before treatment and then decreased to one after treatment. CT, computed tomography; MET, C-11 methionine; PET, positron emission tomography; IG, intracranial germinoma; pre-Tx, pre-treatment; $SUV_{max}$, $SUV_{max}$, maximum standardized uptake value; T/N, tumor-to-normal tissue; post-Tx, post-treatment; SUV, standardized uptake value.

identified between the basal ganglia and other tumor locations. The size of basal ganglia tumors was an average of 28.6 mm with a median of 30 mm. This was relatively large compared to tumors located in other regions, which averaged 19.8 mm with a median of 18 mm. However, tumor MET uptake of basal ganglia tumors was relatively lower than others.

Significant differences in tumor MET uptake were identified according to clinical variables (Table 4). Pre-Tx T/N ratio differed significantly according to sex (P = 0.023), short stature (P = 0.012), and the presence of one-sided weakness and clumsiness (P = 0.005). Pre-Tx SUV-max also differed according to sex (P = 0.001) and one-sided weakness and clumsiness (P = 0.001). In this study, eight of 21 patients (8/21, 38.1%) underwent NGS, and 11 gene

**Table 3. Significant correlations between PET parameters and HCG.**

| PET parameter | Tumor marker | P value |
|---|---|---|
| Pre-Tx T/N ratio | Pre-Tx serum HCG | 0.031[a] |
| Δ T/N ratio | Δ serum HCG | 0.014[b] |
| | Δ CSF HCG | 0.010[b] |
| Δ SUV$_{max}$ | Δ CSF HCG | 0.048[b] |

PET, positron emission tomography; pre-Tx, pre-treatment; T/N, tumor-to-normal tissue; HCG, β-human chorionic gonadotropin; Δ, difference between pre-treatment and post-treatment; CSF, cerebrospinal fluid; SUV$_{max}$, maximum standardized uptake value.

[a]According to Spearman correlation.

[b]According to Pearson correlation.

variants were found. Of the 11 gene variants, a significant difference in PET parameter was identified according to the presence of *KRAS* mutation. Three out of eight patients (3/8, 37.5%) who underwent NGS were identified as the presence of *KRAS* variant. The pre-Tx SUV$_{max}$ of patients with *KRAS* variant was significantly higher than that of patients without *KRAS* variant (P = 0.036).

## MET uptake of IGs and prognosis

In the present study, only one out of 21 IG patients (1/21, 4.8%) died after relapse. There was no significant association between PET parameters and prognosis. Univariate analysis showed that nausea, vomiting (P = 0.034) and dizziness (P = 0.034) were independent predictive factors associated with decreased PFS. Neasea, vomiting (P = 0.034) and dizziness (P = 0.034) were also independent predictive factors associated with decreased OS. In multivariate analysis, there were no significant predictors associated with PFS or OS.

## Discussion

We demonstrated that IGs are well delineated on MET PET and higher MET uptake is exhibited in these tumors compared with INGs in similar locations. These findings support that this

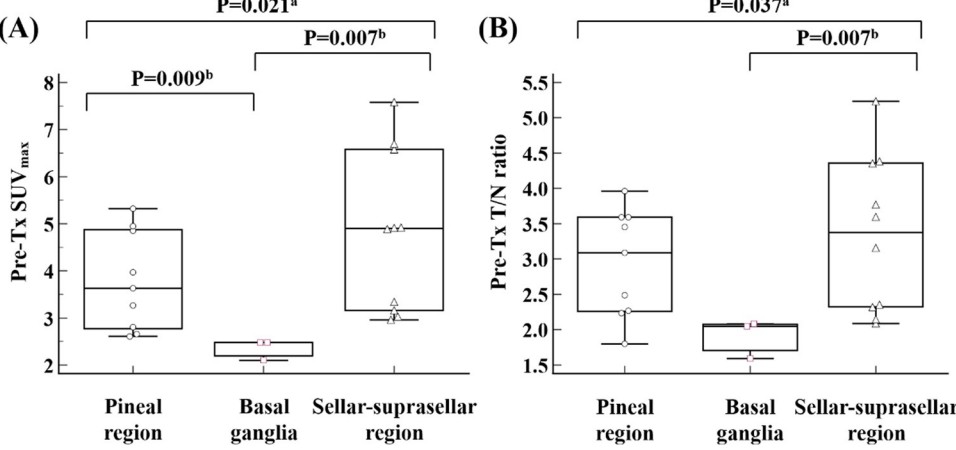

**Fig 3.** Comparisons of (A) pre-Tx SUV$_{max}$ and (B) pre-Tx T/N ratio between IGs located in pineal region, basal ganglia, and sellar-suparsellar region. Pre-Tx, pre-treatment; SUV$_{max}$, maximum standardized uptake value; T/N, tumor-to-normal tissue, IG, intracranial germinoma. [a]According to Kruskal-Wallis test. [b]According to post hoc analysis using Mann-Whitney U test with Bonferroni's method.

**Table 4. Significant differences in PET parameters according to other clinical variables.**

| PET parameter | Clinical variable | P value |
|---|---|---|
| Pre-Tx T/N ratio | Sex | 0.023[a] |
| | Short stature | 0.012[a] |
| | One-sided weakness, clumsiness | 0.005[b] |
| Pre-Tx SUV$_{max}$ | Sex | 0.001[a] |
| | One-sided weakness, clumsiness | 0.001[b] |
| | KRAS variant[c] | 0.036[b] |

PET, positron emission tomography; pre-Tx, pre-treatment; T/N, tumor-to-normal tissue; SUV$_{max}$, maximum standardized uptake value.

[a]According to Independent T-test.

[b]According to Mann-Whitney U test.

[c]Of the 21 patients with intracranial germinoma, only eight underwent NGS.

modality may have a diagnostic role in patients suspected of having IGs, especially for localizing tumors and guiding biopsy. In addition, we identified significant associations between tumor MET uptake and clinical variables. MET PET parameters showed significant associations with tumor location, sex, symptoms, KRAS variant, and HCG.

With conventional neuroimaging such as MRI, early diagnosis of IGs may be difficult due to their slow progression, subtle signal changes, or the absence of a space-occupying lesion, particularly when the tumor is in the basal ganglia [6, 10, 13]. Many different types of tumor can occur in the sellar-suparsellar region, and all of which show similar morphological findings and signal changes on MRI. Therefore, the differential diagnosis of sellar-suprasellar tumors is frequently difficult [9]. Moreover, residual abnormalities caused by tissue necrosis after treatment occasionally mimic residual or recurrent tumor in basal ganglia germinomas [10]. Difficulty diagnosing and evaluating the treatment response may lead to poor prognosis [10]. Some difficult cases of IG cannot be diagnosed by conventional MRI, in these cases, PET imaging could be considered [9]. In a previous study of 10 patients with IGs, the IGs could be detected by FDG PET, but tumor-to-background contrast was poor due to FDG uptake by the nearby normal brain tissue [9]. In most of all cases, the FDG uptake of the tumor was intermediate between FDG uptake of the normal white matter and normal gray matter. On the other hand, almost all of IGs could be visualized on MET PET, therefore, MET is considered to be a good radiotracer for detecting IG. In most of all cases, the IGs showed intense MET uptake than the normal white matter and gray matter [9]. Compared to FDG PET, IGs showed relatively low SUV$_{max}$ and relatively high T/N ratios in MET PET [9]. For these reasons, MET PET is considered useful for tumor localization, tumor contouring, and treatment planning. Previous studies and case reports in patients with IG using MET PET have mainly done research on basal ganglia germinomas and described that MET PET is useful for detection of basal ganglia germinoma without mass effects, differential diagnosis of patients with progressive hemiparesis and hemiatrophy on MRI, assessment of treatment response and residual tumor, monitoring for tumor recurrence, and locating a precise biopsy target [3, 6, 7, 10, 11, 14]. In another previous study in IGs, it elucidated the characteristic MET and FDG PET findings of 10 IGs that develop in basal ganglia, pineal region, and suprasellar region [9]. Unlike in previous studies, the present study conducted research on MET PET findings in more patients with IG, and also analyzed the association between MET parameters and clinical variables including tumor makers, tumor locations, symptom, and NGS data. In addition, in the present

study, the MET PET parameters of IGs and INGs were compared, and the prognosis of IG was also investigated.

In all cases in the present study, the tumors showed markedly higher MET uptake than normal white matter and gray matter tissue. Prior to treatment, all tumors were visual grade 3, confirming that MET PET was useful for detection of IGs. This result is in line with a previous study in that the tumors were well delineated on MET PET [9]. We calculated mean T/N ratios for each visual grade to identify how different each visual grade is quantitatively. The mean T/N ratio of visual grade 3 was 2.98, and the mean T/N ratios of visual grade 2 and 1 were 1.72 and 1.06, respectively. This means that visual grade 3 was approximately three times higher than background, and visual grade 2 was approximately lower than twice the background. It was also confirmed that visual grade 1 was similar to background. In addition, when we compared pre-Tx $SUV_{max}$ and pre-Tx T/N ratio among IGs and INGs located in the pineal region, sellar-suprasellar region, and basal ganglia, $SUV_{max}$ and T/N ratio were higher in IGs. Statistically significant difference was identified in pre-Tx $SUV_{max}$ (P = 0.005), but no statistically significant differences was identified in pre-Tx T/N ratio (P = 0.222). Furthermore, significant differences in tumor MET avidity were identified according to location. MET PET parameters of basal ganglia germinomas were relatively smaller than those of pineal germinoma and sellar-suprasellar germinomas, while no significant differences were identified between sellar-suprasellar tumors and pineal tumors. These results may be related to whether the patients underwent biopsy prior to pre-Tx MET PET/CT. Of 22 IGs, 17 (77.3%) were biopsy performed prior to pre-Tx MET PET/CT, while the remaining five (22.7%) were pre-Tx MET PET/CT followed by biopsy. Previous studies reported that inflammation may occur due to biopsies in tumors [15, 16], and MET uptake may increase due to the inflammation [17–19]. In all three basal ganglia germinomas in this study, biopsy was not performed prior to pre-Tx MET PET/ CT. On the other hand, in 17 (89.5%) of 19 pineal and sellar-suprasellar germinomas, biopsies were performed prior to pre-Tx MET PET/CT. In addition, in this study, mean T/N ratio was high at 3.13 when biopsies were performed prior to pre-Tx MET PET/CT, whereas mean T/N ratio was low at 2.49 when biopsy was not performed prior to pre-Tx MET PET/CT. Therefore, one of the reasons why MET avidity of pineal and sellar-suprasellar germinomas were higher than that of basal ganglia germinomas is possibility of biopsy-induced inflammation. However, in order to confirm this, future MET PET studies in more patients with IG are needed.

Tumor MET uptake was significantly correlated with HCG, but no significant correlation was identified with AFP. GCT can secrete measurable oncoproteins such as HCG and AFP in serum and CSF [4]. Thus, HCG or AFP secretion has been used as a diagnostic marker for intracranial GCT [4, 20]. However, in pure germinomas, only HCG is secreted while AFP levels are normal in serum and CSF [4]. In the present study, the 22 pathologically confirmed IGs consisted of 18 pure germinomas (81.8%), three germinomas with granulomatous reaction (13.6%), and one germinoma with syncytiotrophoblasts (4.6%). This was consistent with previous studies of the physiology of germinoma, suggesting that MET parameters are potential surrogate markers for HCG in IGs.

In this study, tumor MET uptake also varied significantly according to sex. Women had a significantly higher pre-Tx T/N ratio and pre-Tx $SUV_{max}$ than men. It is unclear why the sexual difference in tumor MET uptake exist. There is no obvious elucidation for it [21, 22]. It might be related with male predominance of basal ganglia germinoma. As in previous studies in which basal ganglia germinomas had male predominance [21, 22], all three basal ganglia germinomas were observed in men in this study. The lower tumor MET uptake in men might be due to the higher proportion of basal ganglia tumor in men. However, despite excepting three basal ganglia germinomas, significant difference was observed in pre-Tx $SUV_{max}$

(P = 0.006), and marginal difference was also observed in pre-Tx T/N ratio (P = 0.064) between men and women in this study.

Signs and symptoms are mainly influenced by tumor location, tumor size, extent of tumor involvement, and type of endocrine dysfunction [4]. In this study, there were significant differences in MET PET parameters depending on the presence or absence of these symptoms. Significant differences were identified according to short stature and one-sided weakness, clumsiness. Patients with primary tumors in the sellar-suprasellar region present with a prolonged history of polyuria, polydipsia, growth failure, hypocortisolism, precocious puberty, and hypothyroidism [4]. Patients with acquired endocrine deficiency due to sellar-suprasellar disease usually do not recuperate completely and are dependent on hormone replacement therapy for the rest of their life [4]. In this study, all sellar-suprasellar tumor-bearing patients had symptoms and signs, and three had short stature. Patients with short stature exhibited significantly different PET parameters from those without short stature. Patients with basal ganglia germinomas frequently present with gradually progressive cognitive impairment and hemiparesis [3]. In this study, all patients with one-sided weakness and clumsiness had a tumor located in the basal ganglia. Unlike sellar-suprasellar tumors, as mentioned earlier, basal ganglia tumors have a relatively larger size than others despite relatively low MET uptake. Considering that tumor tracer uptake generally tends to be proportional to tumor size, this result is quite unexpected and the reason is unclear. It will be necessary to confirm this finding in further studies.

In addition, a significant difference in pre-Tx $SUV_{max}$ between patients with *KRAS* variant and patients without *KRAS* variant was observed (P = 0.036). In CNS GCTs, exome sequencing revealed that *KRAS* and *KIT* pathways are frequently mutated [4]. *KRAS* is one of a group of genes involved in the complex signaling pathway called the epidermal growth factor receptor pathway, which regulates cell growth, division, survival, and death [23]. MET reflects proliferative activity [24]. Therefore, an association between MET uptake and the *KRAS* variant is reasonable. Pre-Tx $SUV_{max}$ of IGs is a potential surrogate marker for *KRAS* mutation in pediatric brain tumors.

In the present study, MET avidity of tumors did not have significant prognostic value. This was probably because the treatment outcomes of IGs are highly favorable [4]. Long-term survival rates between 79% and 90% have routinely been attained [25]. In our study, only one of 21 patients (4.8%) relapsed after treatment and died. Because of the excellent prognosis of IGs, the size of the present study was insufficient to assess the discriminative prognostic value of MET parameters.

In previous studies, MET and FDG were mainly used as PET radiotracers in patients with IG. However, in brain imaging, FDG is known to have worse tumor-to-background contrast than MET [9]. In case of MET, the use of MET in medical centers without an on-site cyclotron is restricted because MET has a short radioactivity half-life (20.4 min) [26]. For easy and accurate PET imaging of IGs in more medical centers, future studies are needed for PET radiotracers with better image contrast than FDG and longer radioactivity half-life than MET.

This study has some limitations. First, this was a single-center study with a small number of patients and a retrospective design. Therefore, our results should be verified through future multi-center study involving a large number of patients with a prospective design. Second, in 16 patients (72.2%), biopsies were performed prior to pre-Tx MET PET/CT, and it is possible that the MET parameters of tumor are accompanied by biopsy-induced uptake. Third, MET uptake may be underestimated due to partial volume effect of small lesions less than 2–3 cm in diameter [27]. In the present study, the mean size of IGs was 21 mm (median 18 mm), and several lesions could be affected by partial volume effect.

## Conclusion

MET PET/CT has diagnostic value in patients with IGs. In addition, MET avidity is a potential surrogate biomarker of HCG, which has been used as a diagnostic marker for IGs. Tumor MET parameters also had significant differences according to tumor locations, sex, symptoms, and *KRAS* mutation. However, MET avidity of tumors had no significant prognostic value.

## Supporting information

**S1 Table. Tumor location, $SUV_{max}$, and T/N ratio of pathologically confirmed eight INGs located in pineal region, sellar-suprasellar region, and basal ganglia.**
(DOCX)

**S1 Fig.** Comparisons of (A) pre-Tx SUVmax and (B) pre-Tx T/N ratio between 22 IGs and eight INGs located in pineal gland, pituitary gland, and basal ganglia during the same study period.
(DOCX)

## Author Contributions

**Conceptualization:** Yong-Jin Park, Ji Won Lee, Hee Won Cho, Kyung-Han Lee, Joon Young Choi, Ki Woong Sung, Seung Hwan Moon.

**Data curation:** Yong-Jin Park, Ki Woong Sung, Seung Hwan Moon.

**Formal analysis:** Yong-Jin Park, Ji Won Lee, Hee Won Cho, Yearn Seong Choe, Kyung-Han Lee, Joon Young Choi, Ki Woong Sung, Seung Hwan Moon.

**Funding acquisition:** Yong-Jin Park, Seung Hwan Moon.

**Investigation:** Yong-Jin Park, Ki Woong Sung, Seung Hwan Moon.

**Methodology:** Yong-Jin Park, Seung Hwan Moon.

**Project administration:** Ki Woong Sung, Seung Hwan Moon.

**Resources:** Ji Won Lee, Hee Won Cho, Yearn Seong Choe, Kyung-Han Lee, Joon Young Choi, Ki Woong Sung, Seung Hwan Moon.

**Software:** Yong-Jin Park, Seung Hwan Moon.

**Supervision:** Yearn Seong Choe, Kyung-Han Lee, Joon Young Choi, Ki Woong Sung, Seung Hwan Moon.

**Validation:** Yong-Jin Park, Ki Woong Sung, Seung Hwan Moon.

**Visualization:** Yong-Jin Park, Seung Hwan Moon.

**Writing – original draft:** Yong-Jin Park.

**Writing – review & editing:** Yong-Jin Park, Ji Won Lee, Hee Won Cho, Yearn Seong Choe, Kyung-Han Lee, Joon Young Choi, Ki Woong Sung, Seung Hwan Moon.

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
