## [Decision Letter · Decision Letter 0]

28 Dec 2021

PONE-D-21-34801Value of C-11 methionine PET/CT in patients with intracranial germinomaPLOS ONE

Dear Dr. Moon,

Thank you for submitting your manuscript to PLOS ONE. After careful consideration, we feel that it has merit but does not fully meet PLOS ONE’s publication criteria as it currently stands. Therefore, we invite you to submit a revised version of the manuscript that addresses the points raised during the review process.

ACADEMIC EDITOR:

The article is of scientific interest and original to evaluate the role of [11C]methionine PET/CT in patients with intracranial germinoma (IG). 

Please follow the reviewers comments, with particular attention to the semiquantification analysis reporting SUV values and the improvement of the scientific language. Any significant similarity or difference with other similar article should be discussed more estensively as the limitations of the study.

We look forward to receiving your revised manuscript.

Kind regards,

Pierpaolo Alongi

Academic Editor

PLOS ONE

Additional Editor Comments:

The article is of scientific interest and original to evaluate the role of [11C]methionine PET/CT in patients with intracranial germinoma (IG).

Please follow the reviewer's comments, with particular attention to the semiquantification analysis reporting SUV values and the improvement of the scientific language. Any significant similarity or difference with other similar article should be discussed more estensively as the limitations of the study.

Reviewers' comments:

Reviewer's Responses to Questions

**Comments to the Author**

1. Is the manuscript technically sound, and do the data support the conclusions?

Reviewer #1: Yes

Reviewer #2: Partly

2. Has the statistical analysis been performed appropriately and rigorously? 

Reviewer #1: Yes

Reviewer #2: Yes

3. Have the authors made all data underlying the findings in their manuscript fully available?

Reviewer #1: Yes

Reviewer #2: Yes

4. Is the manuscript presented in an intelligible fashion and written in standard English?

Reviewer #1: Yes

Reviewer #2: Yes

5. Review Comments to the Author

Reviewer #1: This is a very well written paper and deserves only minor changes.

The discussion should be implemented with a comparison with FDG.

The figure legend of fig. 1 should be expanded including also the corresponding T/N ratios and tumor sizes.

Both images should incorporate a color bar.

Reviewer #2: The manuscript aims to evaluate the role of [11C]methionine PET/CT in patients with intracranial germinoma (IG). The topic is certainly of interest to readers, as the literature on it is scarce, but the paper needs to be profoundly improved before being considered for publication.

First of all, the language used should be modified in some cases as it does not sound very "scientific" (for example, page 3 line 54 "situation" could be substitute by "clinical scenario" or something similar).

The introduction section is bare, and many sentences should be further explained (e.g. Why are the tumors occurring at the basal ganglia not easy to distinguish? Or what is the usefulness of MET PET already demonstrated by previous studies?).

In the "Materials and methods"section it might be interesting to add the days between biopsy and PET/CT; do you think the uptake could be influenced by a recent biopsy, bearing in mind that the tracer accumulates where inflammatory changes occur? Furthermore, what does "MET uptake was markedly higher than background" mean? Did you measure the SUV? If so, how many times did the SUV of the lesion have to be compared to that of BKG to be defined as "markedly higher"?

As a final consideration, the sentences regarding study limitations and conclusions are too short and should be improved and the discussion might also include what the future of IG imaging may be, as 11C- needs an on-site cyclotron.

6. PLOS authors have the option to publish the peer review history of their article (what does this mean?). If published, this will include your full peer review and any attached files.

Reviewer #1: **Yes: **Natale Quartuccio

Reviewer #2: No

---

## [Author Response · Author response to Decision Letter 0]

14 Jan 2022

We would like to thank the reviewers and the editor for the careful review of our manuscript.

---

## [Decision Letter · Decision Letter 1]

25 Jan 2022

Value of C-11 methionine PET/CT in patients with intracranial germinoma

PONE-D-21-34801R1

Dear Dr. Moon,

We’re pleased to inform you that your manuscript has been judged scientifically suitable for publication and will be formally accepted for publication once it meets all outstanding technical requirements.

Kind regards,

Pierpaolo Alongi

Academic Editor

PLOS ONE

Additional Editor Comments (optional):

The manuscript has been significantly improved now and can be considered for publication on PLOS ONE.

Reviewers' comments:

Reviewer's Responses to Questions

**Comments to the Author**

1. If the authors have adequately addressed your comments raised in a previous round of review and you feel that this manuscript is now acceptable for publication, you may indicate that here to bypass the “Comments to the Author” section, enter your conflict of interest statement in the “Confidential to Editor” section, and submit your "Accept" recommendation.

Reviewer #1: All comments have been addressed

Reviewer #2: All comments have been addressed

2. Is the manuscript technically sound, and do the data support the conclusions?

Reviewer #1: Yes

Reviewer #2: Yes

3. Has the statistical analysis been performed appropriately and rigorously? 

Reviewer #1: Yes

Reviewer #2: Yes

4. Have the authors made all data underlying the findings in their manuscript fully available?

Reviewer #1: Yes

Reviewer #2: Yes

5. Is the manuscript presented in an intelligible fashion and written in standard English?

Reviewer #1: Yes

Reviewer #2: Yes

6. Review Comments to the Author

Reviewer #1: All the comments have been addressed correctly by the authors. I have no further concerns about the manuscript.

Reviewer #2: The manuscript has been significantly improved now and can be considered for publication on PLOS ONE.

7. PLOS authors have the option to publish the peer review history of their article (what does this mean?). If published, this will include your full peer review and any attached files.

Reviewer #1: **Yes: **Natale Quartuccio

Reviewer #2: **Yes: **Priscilla Guglielmo

---

## [Editor Report · Acceptance letter]

28 Jan 2022

PONE-D-21-34801R1 

Value of C-11 methionine PET/CT in patients with intracranial germinoma 

Dear Dr. Moon:

I'm pleased to inform you that your manuscript has been deemed suitable for publication in PLOS ONE. Congratulations! Your manuscript is now with our production department. 

Kind regards, 

on behalf of

Dr. Pierpaolo Alongi 

Academic Editor

PLOS ONE